# Quantitation of a Urinary Profile of Biomarkers in Gaucher Disease Type 1 Patients Using Tandem Mass Spectrometry

**DOI:** 10.3390/diagnostics12061414

**Published:** 2022-06-08

**Authors:** Iskren Menkovic, Michel Boutin, Abdulfatah Alayoubi, Filipa Curado, Peter Bauer, François E. Mercier, Christiane Auray-Blais

**Affiliations:** 1Division of Medical Genetics, Department of Pediatrics, Centre de Recherche-CHUS, Faculty of Medicine and Health Sciences, Université de Sherbrooke, CIUSSS de l’Estrie-CHUS, 3001, 12th Avenue North, Sherbrooke, QC J1H 5N4, Canada; iskren.menkovic@usherbrooke.ca (I.M.); michel.boutin2@usherbrooke.ca (M.B.); 2Divisions of Experimental Medicine and Hematology, Department of Medicine, Faculty of Medicine, McGill University, Lady Davis Institute for Medical Research, Jewish General Hospital, 3755, Côte Sainte-Catherine, Montreal, QC H3T 1E2, Canada; abdu.alayoubi@gmail.com (A.A.); francois.mercier@mcgill.ca (F.E.M.); 3Department of Biochemistry and Molecular Medicine, College of Medicine, Taibah University, University Road, Madinah 42353, Saudi Arabia; 4CENTOGENE GmbH, 18055 Rostock, Germany; filipa.curado@centogene.com (F.C.); peter.bauer@centogene.com (P.B.)

**Keywords:** Gaucher disease type 1, urine, biomarkers, glucosylsphingosine, lyso-Gb_1_, lyso-Gb_1_ analogs, polycyclic lyso-Gb_1_ analogs, tandem mass spectrometry, ultra-performance liquid chromatography

## Abstract

Gaucher disease is a rare inherited disorder caused by a deficiency of the lysosomal acid beta-glucocerebrosidase enzyme. Metabolomic studies by our group targeted several new potential urinary biomarkers. Apart from lyso-Gb_1_, these studies highlighted lyso-Gb_1_ analogs −28, −26, −12 (A/B), +2, +14, +16 (A/B), +30, and +32 Da, and polycyclic lyso-Gb_1_ analogs 362, 366, 390, and 394 Da. The main objective of the current study was to develop and validate a robust UPLC-MS/MS method to study the urine distribution of these biomarkers in patients. Method: Urine samples were purified using solid-phase extraction. A 12 min UPLC-MS/MS method was developed. Results: Validation assays revealed high precision and accuracy for creatinine and lyso-Gb_1_. Most lyso-Gb_1_ analogs had good recovery rates and high intra- and interday precision assays. Biomarker-estimated LOD and LOQ levels ranged from 56–109 pM to 186–354 pM, respectively. Comparison between GD patients and healthy controls showed significant differences in most biomarker levels. Typically, treated GD patients presented lower biomarker levels compared to untreated patients. Conclusions: These data suggest that the metabolites investigated might be interesting GD biomarkers. More studies with a larger cohort of patients will be needed to better understand the clinical significance of these GD biomarkers.

## 1. Introduction

Gaucher disease (GD; OMIM 230800, 230900, and 231000) is an autosomal recessive disorder caused by a deficiency in the lysosomal enzyme acid beta-glucocerebrosidase (GCase; OMIM 606463) [1] encoded by the *GBA* gene. This leads to the accumulation of several glycosphingolipid species in numerous organs and tissues, namely the spleen, the liver, and the bone marrow [2]. This ultimately impairs the natural homeostasis of the tissue and results in multisystemic clinical manifestations varying in frequency and severity [3]. While GD shows a wide continuum of phenotypes ranging from asymptomatic to early death, patients are classified into three subtypes based on the severity of the clinical manifestations and degree of neurological involvement [4,5]. GD type 1 (OMIM 230800) is the most common form of the disease and represents approximately 90–95% of all GD cases [6]. Patients typically exhibit visceral manifestations (hepatomegaly and/or splenomegaly), hematological complications (cytopenia) and bone disease. Neurological involvement in this subtype is either mild or absent [6]. GD type 2 represents the most severe form of the disease and is commonly referred to as the acute neuronopathic GD. Normally, in GD type 2, neurological involvement appears in the first six months of life and leads to death by the age of two [7]. In comparison, GD type 3 or the chronic neuronopathic form of the disease may appear months to years after birth. Moreover, in GD type 3, the progression and the severity of neurological involvement are slower and less severe than in GD type 2 [7]. Both GD type 2 and type 3 experience visceral, hematological, and bone disease in addition to neurological complications.

Enzyme replacement therapy (ERT) and substrate reduction therapy (SRT) are some of the treatments available for GD patients. These therapies are efficient for clinical manifestations related to visceral manifestations, hematological complications, and bone disease [8,9]. However, ERT has a limited impact on CNS-related complications since it does not cross the blood–brain barrier [10]. Moreover, it is well documented that the early initiation of treatment reduces the progression of clinical manifestations and improves the outcome [11,12,13]. Unfortunately, due to the important phenotypic variability and wide spectrum of severity associated with the disease, a confirmed diagnosis may, in some cases, be quite challenging [13]. Biomarkers found in the blood such as angiotensin-converting enzyme (ACE), tartrate-resistant acid phosphatase (TRAP), chitotriosidase, and CCL18 can be used for the routine follow-up of patients [14,15,16,17,18]. However, they are not specific to GD [19]. In recent years, lyso-Gb_1_ emerged as a key biomarker for the monitoring and progression of the disease [20]. Moreover, this biomarker could potentially be used for the diagnosis of the disease [21]. Considering the nature of the disease and associated clinical manifestations, the blood matrix was mostly investigated as a source of biomarkers related to the disease, potentially overlooking biomarkers found in other matrices such as urine. Recently, metabolomic studies performed by our research group in plasma and urine highlighted several different metabolites increased in each matrix [22,23]. More specifically, N-palmitoyl-O-phosphocholine serine, sphingosylphosphorylcholine, lyso-Gb_1_ as well as lyso-Gb_1_ analogs −28, −2, +14, and +18 Da were targeted as potential biomarkers in plasma specimens. Meanwhile, lyso-Gb_1_, polycyclic lyso-Gb_1_ analogs 362, 366, 390, and 394 Da in addition to lyso-Gb_1_ analogs −26, −12, +2, +14, +16 (A) and (B), +30, +32, and +44 Da were highlighted as potential biomarkers in urine as shown in Figure 1.

These two metabolomic studies also suggested that the distribution of biomarkers varies significantly from one matrix to the other. Indeed, some lyso-Gb_1_ analogs were more abundant than lyso-Gb_1_ itself in urine [22]. Moreover, analogs +44, +32, +30, +2, and −26 Da were not detected in plasma [23]. Additionally, the metabolomic study performed in urine specimens highlighted a novel class of GD biomarkers called polycyclic lyso-Gb_1_ analogs. These latter biomarkers were over 10 times more elevated than lyso-Gb_1_ itself, as well as the related analogs in some urine specimens [22,24].

In the past, our research group performed several metabolomic studies in urine on Fabry disease, where glycosphingolipid accumulation is also involved. These studies led to the discovery of novel lyso-Gb_3_ analogs with different modifications on the sphingosine moiety [25,26]. Another study highlighted that several lyso-Gb_3_ analogs had a significant correlation with specific clinical manifestations of the disease [27]. Considering that Gaucher disease and Fabry disease have closely related metabolic pathways with structurally similar resulting biomarkers, we hypothesized that some urinary biomarkers observed in GD may also show significant correlations with specific clinical manifestations. Therefore, urinary biomarkers highlighted by the GD metabolomic study need to be further investigated to evaluate correlations with specific clinical manifestations of the disease. Therefore, the main objective of this research project was to develop and validate a robust tandem mass spectrometry urine quantitation method for lyso-Gb_1_ and related analogs (−26, −12, +2, +14, +16, +30, and +32 Da), as well as polycyclic lyso-Gb_1_ analogs 362, 366, 390, and 390 Da. Upon the validation process, correlations were investigated between biomarker levels and patient-specific clinical manifestations. 

## 2. Materials and Methods

### 2.1. Reagents

Carbon 13 labelled-glucosylsphingosine (lyso-Gb_1_-(^13^C_6_)) powder (98% purity) was acquired from Matreya (Pleasant Gap, PA, USA). Creatinine powder (98% purity) was purchased from Sigma Aldrich (Saint-Louis, MO, USA) while deuterated creatinine (Creatinine-D_3_) standard (98% purity) was obtained from CDN Isotopes Inc. (Pointe-Claire, QC, CAN). Synthetic urine was supplied by BioIVT (Westbury, NY, USA). UPLC grade methanol (MeOH) and acetonitrile (ACN) were purchased from EMD Chemicals Inc. (Darmstadt, Germany). Optima LC/MS grade water and MeOH, as well as American Chemical Society (ACS) grade ammonium formate (Amm. Form.), ammonium hydroxide (NH_4_OH; 28–30% purity), and O-phosphoric acid (H_3_PO_4_; 85%) were obtained from Fisher Scientific (Fair Lawn, NJ, USA). Finally, formic acid (FA; >99%) was obtained from Acros Organics (Morris Plain, NJ, USA).

### 2.2. Ethics Approval

The research project presented in this paper was approved by the Research Ethics Board (REB) of the Faculty of Medicine and Health Sciences of the Centre hospitalier universitaire de Sherbrooke (CHUS) under the project ID MP-31-2017-1414. Jewish General REB also approved the study (REB Project ID MEO-31-2020-1937). Patients recruited from Centogene were part of the Lyso-Prove project, which was approved by Universität Rostock REB under project ID A 2015-0025. Informed consent from all patients and healthy controls was obtained. 

### 2.3. Study Population

Liquid urine samples from treated and untreated GD type 1 patients were collected from different countries, namely Canada, Germany, Israel, and Greece. Males and females, both minors and adults were enrolled in the study. Upon arrival at our facility, samples were kept at −30 °C until analysis. For all patients involved in the study, a marked GCase enzyme deficiency in peripheral blood leukocytes was demonstrated. Moreover, all patients had biallelic mutations of the *GBA* gene which was revealed by sequencing. Healthy controls considered for this study did not have any lysosomal storage diseases (LSDs) or other comorbidities.

### 2.4. Calibration Curves and Quality Control Samples Preparation

The lyso-Gb_1_ powder was dissolved in MeOH to obtain two stock solutions with concentrations of 2000 and 20,000 nM. Using these stock solutions, a 9-point calibration curve solution with respective concentrations of 0, 2, 20, 45, 75, 125, 250, 500, and 800 nM in the urine matrix was prepared. Regarding the preparation of quality control (QC) samples, spiked samples with concentrations of 10, 187, and 650 nM of lyso-Gb_1_ were used for low, medium, and high QC samples. 

For the quantitation of creatinine, calibration curve points with concentrations of 0, 1, 2, 5, 7.5, 10, 15, 20, and 30 mM were prepared in water and used for sample quantitation. Urine samples from healthy controls with creatinine concentrations of 1.9, 9.5, and 18 mM were used as QC samples for creatinine quantitation.

### 2.5. Sample Preparation

Urine samples were thawed at room temperature. Samples were then vortexed for 10 s and a volume of 200 μL was transferred into a 2 mL Eppendorf polypropylene tube. A volume of 500 μL of a 2% H_3_PO_4_ solution was added to the sample, followed by a volume of 500 μL of a MeOH solution containing 10 mM lyso-Gb_1_-(^13^C_6_). Finally, a volume of 10 μL of a solution of 20 mM of creatinine-D_3_ was added to each sample. Samples were then purified using Oasis mixed-mode cation-exchange (MCX) solid-phase extraction (SPE) cartridges (Waters Corp. Milford, MA, USA). Briefly, SPE cartridges were conditioned with 1 mL of MeOH and 1 mL of a 2% H_3_PO_4_ solution. Sample solutions were then vortexed for 5 s and transferred into the cartridges. Loaded cartridges were washed with 1 mL of a 2% formic acid solution, followed by a wash using 1 mL of a 0.2% FA solution prepared in MeOH. The analyte elution step was performed using 600 μL of the 2% ammonium hydroxide solution prepared in MeOH. Samples were evaporated to dryness under a stream of nitrogen, followed by a resuspension in 200 μL of the mobile phase A (94.5:2.5:2.5:0.5 ACN: MeOH: H_2_O: FA + 5 mM Amm. Form). Samples were ultimately transferred into a 300 μL glass insert added to a 2 mL vial. A volume of 4 μL was injected into the UPLC-MS/MS system for analysis.

### 2.6. Instrumentation and Parameters

The quantitation of potential GD biomarkers in urine, as well as creatinine, was performed using an Acquity I-class UPLC system coupled to a Xevo TQ-S tandem mass spectrometer (Waters Corp. Milford, MA, USA). Detailed parameters used are shown in Table 1.

### 2.7. Biomarker Quantitation

Creatinine was quantified using commercially available standard and internal standard. Considering that creatinine is a molecule found in all urine samples (even in synthetic urine), the calibration curve needed to be prepared in water to avoid underestimating creatinine levels in samples. Moreover, to obtain the most accurate and precise quantitation possible, a 9-point calibration curve with concentrations ranging from 0 to 30 mM of creatinine was used. A quadratic curve with a weighing 1/x and the original value included were selected. Data processing was achieved with MassLynx/QuanLynx software version 4.2 SCN982 (Waters Corporation, Milford, MA, USA). Regarding lyso-Gb_1_ and analogs, including polycyclic analogs, quantitation was achieved using a 9-point calibration prepared in synthetic urine with a commercially available lyso-Gb_1_ standard. Considering that no standards or internal standards are available for lyso-Gb_1_ analogs, we performed a relative quantitation of these latter metabolites using the lyso-Gb_1_ calibration curve and internal standard. Similar to the creatinine, lyso-Gb_1_ quantitation was achieved using a quadratic calibration curve with a weighing 1/x. Data were also processed with the MassLynx/QuanLynx software version 4.2 SCN982 (Waters, Milford, MA, USA).

### 2.8. Method Validation

Various concentrations of urine samples were selected for the creatinine assays. The levels were measured using a previously described method [28]. Briefly, the method consists of successive dilution steps of a urine sample in a 65:35 ACN: H_2_O +50 mM Amm. Form and 0.5% FA to achieve a total final dilution factor of 1:4000. A deuterated internal standard (Creatinine-D_3_) was also added to increase the method’s reliability. Samples were then injected into a UPLC-TOF/MS for quantitation. A 9-point calibration curve with concentrations ranging from 1 to 30 mM of creatinine including a blank with the internal standard only was used. UPLC-MS/MS parameters are shown in Appendix A. The selected urine samples had creatinine concentration levels of 1.9 mM (low QC), 9.5 mM (medium QC), and 19.0 mM (high QC). These urine matrices were then spiked with lyso-Gb_1_ standard to obtain QCs with concentrations of 10 nM (low QC), 187 nM (medium QC), and 650 nM (high QC). Using these prepared QC samples, both intraday (5 replicates per QC per day) and interday (over 5 different days) precision and accuracy parameters were evaluated. Considering the limited volume of GD patient urine samples, intraday precision assays (5 samples/day) and interday precision assays (3 samples/day for 5 days) for lyso-Gb_1_ analogs and polycyclic analogs were evaluated using a HQC (patient with the highest concentration of biomarkers in the cohort) and a LQC (patient with average biomarker concentrations). The lyso-Gb_1_ calibration curve correlation factor was evaluated for each validation day using a freshly prepared curve. Sample stability for 24 h, 48 h, one week, and one month at −30 °C, 4 °C, and room temperature (22 °C) was assessed by comparing stability QCs with their respective baseline values. Glass and plasticware adhesion were evaluated by transferring the resuspended samples three times into a vial using either a Pasteur pipette (glassware adhesion) or a regular plastic tip (plastic adhesion). Freeze/thaw cycles biases were evaluated by comparing measured levels of lyso-Gb_1_ from aliquots of a patient sample. Half of these aliquots went subject to three freeze/thaw cycles (*n* = 3) while the other half were prepared following the regular protocol. The matrix effect was measured by post-column infusion using lyso-Gb_1_-(^13^C_6_) at a concentration of 1 µg/mL and a constant flow rate set at 0.15 µL/min. Limits of detection (LOD) and limits of quantification (LOQ) were calculated using the standard deviation obtained by analyzing 10 times the same patient sample. The standard deviation was then multiplied by three to estimate the LOD and by ten to estimate the LOQ.

## 3. Results and Discussion

### 3.1. Chromatographic Separation 

The chromatographic method was developed with the following two main objectives: (1) the separation of lyso-Gb_1_ from its structural isomer galactosylsphingosine (psychosine), and (2) a reduction in the matrix effect in the analyte and internal standard elution regions. When considering the separation of lyso-Gb_1_ from psychosine, some may argue that the psychosine concentration is negligible compared to the concentration of lyso-Gb_1_ in GD patients. We believe that a chromatographic separation of these two structural isomers allows greater applicability of the method described herein. Indeed, the chromatographic separation of lyso-Gb_1_ and psychosine favors the adaptability of the method for biomarker quantitation, particularly for Krabbe disease. Moreover, the efficient separation of both metabolites eliminates the risk of potential contamination from one metabolite to the other, which could affect the accuracy of the results. As shown in Figure 2, the separation of lyso-Gb_1_ and psychosine was achieved using the proposed chromatographic parameters.

As previously mentioned, the chromatographic separation of lyso-Gb_1_ and all related analogs was achieved using an isocratic separation during the elution of all compounds of interest followed by a short linear gradient to wash off the column prior to re-equilibration. The resulting chromatography is shown in Figure 3. 

Regarding the matrix effect, there are no commercially available standards or internal standards for lyso-Gb_1_ analogs or polycyclic analogs. Therefore, lyso-Gb_1_-(^13^C_6_) was used as an internal standard for all these molecules. To correct the matrix effect, all these metabolites need to be compared under the same conditions, targeting a similar matrix effect. A relatively stable matrix effect was achieved for the entire duration of the chromatographic separation for all analytes using an isocratic gradient. As shown in Figure 4, there were no regions with a significant ion enhancement or suppression between 3 and 10 min corresponding to the retention time of lyso-Gb_1_ and the related metabolites. More specifically, a continuous infusion of lyso-Gb_1_-(^13^C_6_) revealed three regions with high ion enhancement effects at 5.39 min, 6.16 min, and 6.74 min. Most analytes do not elute within those three regions; hence, they are not affected by the ion enhancement effect observed. However, lyso-Gb_1_ analogs +14, +16, and +30 Da do have retention times relatively close to 5.39. However, even for those analytes, considering there is some difference in the retention time, we are confident that biases in quantitation caused by the matrix effect will be less than 15% for all analytes.

Regarding the lyso-Gb_1_ analog +14 Da, two peaks were visible in the chromatogram (Figure 3A). To assess if both compounds were indeed lyso-Gb_1_ analogs, fragmentation tests were performed using an Acquity UPLC (Waters Corp., Milford, MA, USA) coupled to a time-of-flight mass spectrometer (Synapt G1, Waters Corp., Milford Massachsetts). The UPLC-MS-TOF parameters used are described in Appendix A, as well as in a previously published paper by our research group [22]. The collision energy ramp used specifically for metabolites discussed herein ranges from 15 to 25 V. These tests reveal that both compounds were indeed isoforms of the same molecule since both metabolites produced similar fragments in almost identical relative abundances (Figure 5). Therefore, both peaks were considered as lyso-Gb_1_ analog +14 Da. 

The metabolomic study performed in GD patient urine specimens revealed the existence of a metabolite with two similar transitions for lyso-Gb_1_ analog +16 Da [22]. More specifically, for these analogs, accurate masses of 478.3357 Da and 478.3008 Da were obtained. As suspected, fragmentation tests performed on these two molecules revealed that they are indeed structurally different [22]. To avoid any confusion when referring to these two metabolites, they were distinguished as lyso-Gb_1_ analog +16 Da (A; 478.3357 Da; C_24_H_48_NO_8_) and (B; 478.3008 Da; C_23_H_44_NO_9_). Therefore, this nomenclature was used thereafter. As shown in Figure 3A, several peaks with retention times from 5.43 to 5.56 min were detected. Fragmentation tests were made to assess if these peaks were structural isomers of analog +16 Da (A) or if they were entirely different molecules as previously noticed with analog +16 Da (B). The UPLC-TOF/MS results are shown in Appendix A. The collision energy ramp used for the fragmentation tests ranged from 15 to 25 V. These fragmentation tests showed the same profile for both peaks and therefore peaks with retention times of [5.43–5.56] min were quantitated as the analog +16 Da (A) (Figure 6). 

Two chromatographic peaks were observed for analog −12 Da as shown in Figure 3A. Both peaks were flagged as potential biomarkers for GD during the metabolomic study performed in urine. These two metabolites had respective accurate masses of 450.2702 Da and 450.3007 Da. 

Unfortunately, due to its limited concentration in samples and the presence of contaminants with similar retention times, it was only possible to find the peak associated with this metabolite with an accurate mass of 450.3007 Da when analyzed by MS/TOF. Since the precursor ion could not be found for the metabolite of 450.2702 Da, it was not possible to perform fragmentation tests for that specific molecule. 

However, the metabolomic study performed previously revealed an increase in GD patients while the metabolite was not detectable in most healthy controls (see Appendix A) [22]. Specifically, there was one healthy control out of 15 who appeared to have an elevation of this metabolite. However, a more in-depth analysis revealed that this increase was caused by a contaminant with a similar mass and retention time. Indeed, the accurate masse measured for the contaminant was 450.3189 Da, which differs from the accurate mass observed for lyso-Gb_1_ analog −12 Da (A; 450.2702 Da) and (B; 450.3007 Da). These masses for lyso-Gb_1_ analog −12 Da suggest an elemental composition of C_21_H_40_NO_9_ and C_22_H_44_NO_8_, for (A) and (B), respectively, both of which are compatible with lyso-Gb_1_ analog potential structures. Additionally, accurate masses and exact masses differ by −0.2 ppm and −13.3 ppm, respectively for metabolites (A) and (B). Additionally, both molecules appear to be greatly increased in patients severely affected, and it decreases upon treatment by ERT. Hence, we have decided to quantify these metabolites individually and refer to them as lyso-Gb_1_ analogs −12 (A; 450.2702 Da) and (B; 450.3007 Da).

A shoulder peak was also visible for lyso-Gb_1_ analogs −26 and +30 Da (Figure 3A). Regarding the analog +30 Da, similar to analog −12 Da, a second metabolite with an accurate mass of 492.3148 Da was flagged during the metabolomic study in urine [22]. This is relatively close to the accurate mass measured for the other, more abundant metabolite (492.3159 Da) and thus does not suggest different empirical formulas for these two molecules. Again, due to its limited concentration level, it was not possible to find a peak associated with this metabolite during fragmentation tests. However, the quantitation of both metabolites using a more sensitive mass spectrometer instrument (Xevo TQ-S, Waters Corp., Milford, MA, USA) showed that the ratio between the two isomers seems relatively constant in samples. We, therefore, decided to include this shoulder peak as part of the analog +30 Da.

Finally, the potential isomer for the analog −26 Da, was not observed during the metabolomic study. However, this marker is present in GD samples only, potentially excluding the hypothesis that it may be a contaminant. Moreover, its concentration appears related to the abundance of the dominant chromatographic peak observed for analog −26 Da. Indeed, in the vast majority of samples analyzed, the ratio between the shoulder peak and the peak itself appears relatively constant. We have therefore decided to include this potential isomer as part of lyso-Gb_1_ analog −26 Da.

### 3.2. Method Validation

The method was validated using urine samples spiked with various concentrations of lyso-Gb_1_. As mentioned previously, there are no commercially available standards or isotopically labelled standards for lyso-Gb_1_ analogs. Therefore, to quantify these molecules, we selected a commercially available standard with similar chemical properties such as the pKa of the ionization sites, and overall similar structures. Considering the closely related chemical and physical properties of lyso-Gb_1_ and all related analogs, these latter compounds were quantified using the lyso-Gb_1_ standard. Therefore, the method of validation was performed for all lyso-Gb_1_ related compounds using lyso-Gb_1_ standards, as we expected similar results for both types of molecules. Results obtained for each validation parameter are shown in Table 2. Limits of detection and limits of quantitation for lyso-Gb_1_, lyso-Gb_1_ analogs and polycyclic analogs are shown in Table 3. Detailed results regarding intraday and interday precision and accuracy results for QCs and the stabilities of all metabolites are shown in Appendix A. Considering that creatinine may be quite abundant in some urine samples and that an isotopically labelled internal standard (creatinine-D_3_) was used, parameters such as the limit of quantification (LOQ), the limit of detection (LOD), and the recovery were not evaluated. The adhesion of lyso-Gb_1_ analogs and polycyclic analogs to glass- and plasticware was not significant considering that the variations from baseline values ranged from −3.7% to 1.2%.

### 3.3. Biomarker Quantitation

Patients and controls were separated into three different groups, namely untreated GD patients (*n* = 18), treated GD patients (*n* = 9), and healthy controls (*n* = 9) as shown in Figure 7. Patients in the treated group were under ERT for at least six months.

All untreated GD patients but two had a disease severity that could be qualified as “borderline to mild” based on the GD type 1 severity scoring system proposed by Weinreb et al. in 2010. The patients (*n* = 3) who were classified as “moderate or marked” with a score ranging from 3–9, did show a higher level of biomarkers, especially for analogs +32, +30, +16 (B), +2 Da, as well as polycyclic analogs 394, and 366 Da. Unfortunately, considering the limited number of patients affected by a more severe form of the disease in our cohort, it is not possible to confirm if these markers have a significant correlation with the overall disease severity.

Biomarker quantitation showed that all of the proposed biomarkers were either not detected or below the limit of quantification in healthy controls. As expected, and confirmed by the Mann–Whitney U test, all biomarkers under study had a statistically significant difference in both GD groups (treated and untreated) from the control group at *p* < 0.01. Regarding treated and untreated patients, lyso-Gb_1,_ as well as lyso-Gb_1_ analogs +32, +14, −28 Da, and polycyclic analog 394 Da, were the only biomarkers that showed a statistically significant difference between treated and untreated patients at *p* < 0.01. The *p*-values associated with each group are summarized in Appendix A. While these data suggest that the aforementioned metabolites may be more sensitive for patient follow-up and monitoring, we would like to emphasize that most untreated patients in our cohort had an attenuated form of the disease. Based on studies performed on other LSDs, it was demonstrated that even after treatment with ERT, residual levels of biomarkers are still present in tissues and biological fluids [26]. Patients classified as “borderline to mild” typically do not have an important concentration of biomarkers, whereas for patients who are treated, biomarkers levels never truly return to baseline values such as those observed in healthy controls. This might be the reason why only a few biomarkers had a statistically significant difference between treated and untreated patients. A larger cohort with a wider range of disease severity would be required to fully investigate this hypothesis and to confirm if some metabolites are more reliable for the follow-up and monitoring of patients than others. Out of the 18 untreated patients in our cohort, only two were characterized as having moderate disease severity while the rest were characterized as having a very attenuated form of the disease. More specifically, these patients displayed numerous clinical manifestations such as bone pain, Erlenmeyer flask deformity, infarction, and avascular necrosis. Cytopenia and organomegaly were also observed. Both patients did have a bone marrow burden score of between six and eight for both the lower limb and spine. These patients also had significantly higher concentration levels of most biomarkers than other patients classified as “borderline to mild”. In fact, out of the 14 potential biomarkers under study, only two, namely polycyclic analogs 366 and 394 Da were not significantly more elevated in these two patients compared to other GD patients affected by a less severe form of the disease. Unfortunately, while these data suggest that lyso-Gb_1_ and most analogs could correlate with disease severity, it is not possible, at this point, to confirm which metabolite has a significant correlation with the severity of GD. A larger cohort of untreated patients with a wider range of severity will be needed to fully investigate this.

### 3.4. Biomarker Concentrations upon Treatment

Three patients in our cohort had an extended follow-up (15 months, 6 timepoints) post-ERT. As expected, a decreasing trend in biomarker levels was seen following treatment (Figure 8). Clinical data regarding organomegaly and cytopenia associated with each time point are shown in Table 4.

While not always linear, a decreasing trend was observed for most biomarkers 15 months after treatment. The biomarker variation for two other patients with a longitudinal follow-up also showed a similar phenomenon, with an overall decreasing trend but not necessarily in a constant linear fashion (Appendix A). Studies performed on other diseases tend to demonstrate a certain variability in urinary biomarker concentration levels. For Fabry disease, the authors hypothesized that biomarker level variability can be related to physical activity, food intake, and water consumption [29]. Regarding other treated patients in our cohort without a longitudinal follow-up (*n* = 20), a decrease in biomarker levels 3 months post-ERT was noticed. However, more data from treated patients affected by a wider range of severity will be needed to assess the reliability of the biomarkers studied for follow-up purposes. Indeed, considering that our cohort was mostly from patients affected by a less severe form of the disease, a decrease in biomarkers may have not been as noticeable as in a patient with a more severe form of the disease with higher concentration levels of biomarkers.

Moreover, although the plasma quantitation of biomarkers was found to be a reliable method for the follow-up of patients, we believe that urine specimens might also be helpful for biomarker evaluation as part of a biochemical profile for GD patients. Indeed, urine collection is much less invasive than blood collection. Moreover, collection techniques could potentially be developed where patients would use a filter paper to collect a urine sample at home and send it by regular mail to the appropriate facility. This would facilitate the follow-up of patients without requiring the patient to go to the hospital or collection centers. In fact, filter paper methods have already been developed for other LSDs such as Fabry disease and mucopolysaccharidoses [30,31,32,33].

## 4. Conclusions

A metabolomic study allowed us to identify and elucidate the chemical structure of these new potential metabolites for GD patients [22]. In this latter study, we used a UPLC-QTOF, which has a better mass accuracy but is significantly less sensitive for targeted analyses than the triple quadrupole mass spectrometer used in the current manuscript. The gain in precision during the metabolomic study was a valuable tool for the structural elucidation process of the highlighted metabolites. In comparison, the focus of this current manuscript was to develop a sensitive and rapid method to quantify the metabolites highlighted in the metabolomic study. Following the detection of potential GD biomarkers in urine, we developed and validated a novel UPLC-MS/MS method to perform the quantitation of these metabolites of interest. Once validated using available commercial standards, this method aimed to investigate potential correlations between biomarker levels and clinical manifestations related to GD. Our data revealed that all biomarkers under study were significantly elevated in GD patients compared to healthy controls. A significant decrease in most biomarkers was noticed at the three or six months visit post-treatment. Although these preliminary results are noteworthy, a larger cohort with patients with severe clinical manifestations is warranted to fully investigate the correlations between biomarker concentration levels and the severity of the disease displayed by patients. Regarding the three patients having longitudinal monitored treatment by ERT, a decreasing trend (although with variability), in most biomarker concentrations in a relatively short time (15 months) was observed, corresponding to an improvement in organomegaly and hematologic manifestations. In conclusion, we strongly suggest the evaluation of a profile of various types of GD biomarkers (e.g., biochemical and imaging) for the efficient monitoring and follow-up of patients.

## 5. Future Perspectives

The analysis of a larger cohort will be required to study the correlation between biomarker concentration levels and clinical manifestations of GD. Furthermore, a comparison between plasma biomarkers previously studied and the biomarkers discussed herein is needed to assess the strengths and weaknesses of both methods. Finally, different sampling collection procedures, such as urine collected on filter paper, will be investigated in the near future considering the advantages for shipment and sample storage compared to “liquid” urine specimens.

## 6. Study Limitations

The lack of a standard and internal standard is a limitation for polycyclic analog quantification. The use of more appropriate standards would surely improve the accuracy of the method regarding polycyclic analogs as these new standards would better reflect the ionization potential of these novel metabolites and would possibly correct the variability related to the sample preparation procedure. Another limitation in this study is the narrow range of severity of GD patients recruited in our cohort. This has limited our ability to investigate potential correlations between biomarker concentration levels and clinical manifestations of the disease which would have increased the statistical power of the study.

## Figures and Tables

**Figure 1 diagnostics-12-01414-f001:**
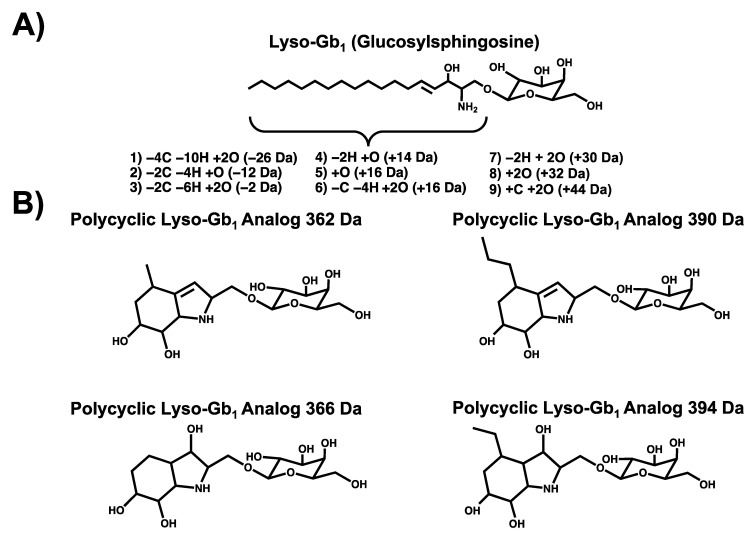
Potential GD biomarkers highlighted in urine specimens: (**A**) lyso-Gb_1_ chemical structure with structural modifications observed on the sphingosine moiety in lyso-Gb_1_ analogs; (**B**) Polycyclic lyso-Gb_1_ analog chemical structures [22].

**Figure 2 diagnostics-12-01414-f002:**
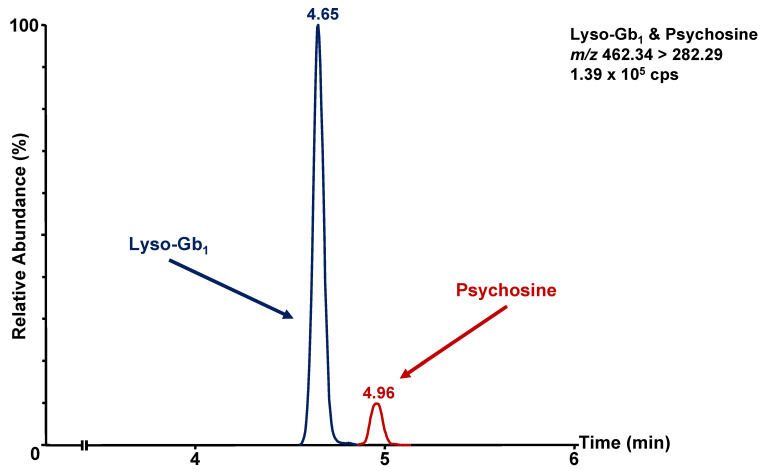
Ion chromatogram obtained by multiple reaction monitoring (MRM) of 60 ng on-column of lyso-Gb_1_ standard (blue, retention time: 4.65 min) and 10 ng on-column of psychosine standard (red, retention time: 4.96 min).

**Figure 3 diagnostics-12-01414-f003:**
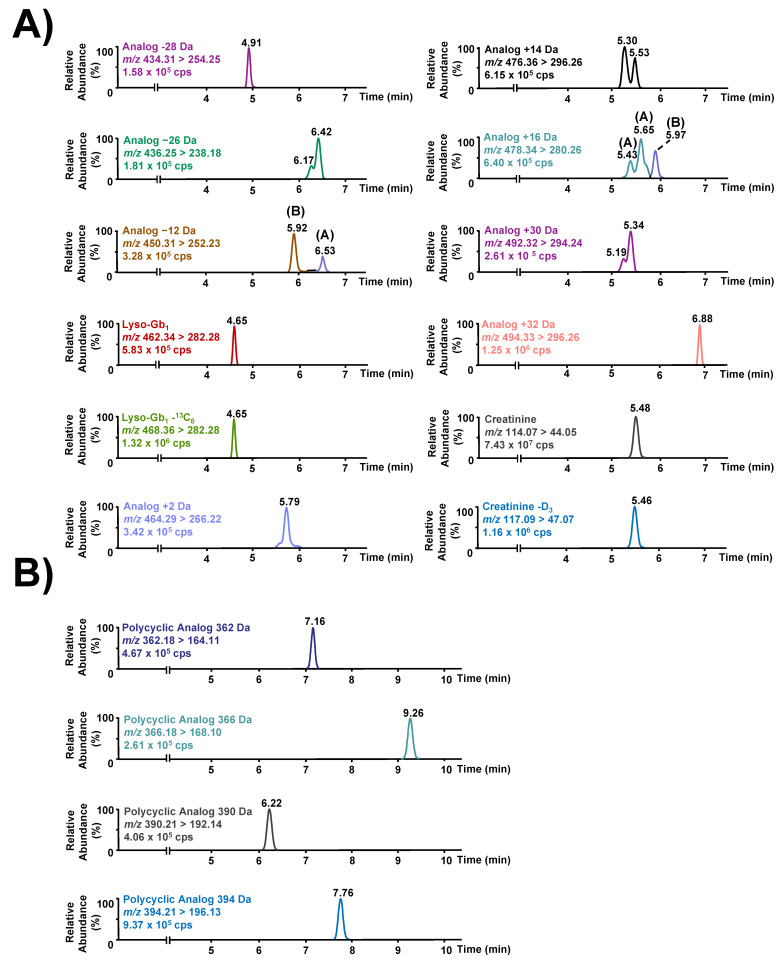
Chromatographic separation of potential lyso-Gb_1_ biomarkers and creatinine with their respective internal standards. (**A**) Ion chromatograms using a MRM mode for lyso-Gb_1_, lyso-Gb_1_-(^13^C_6_), lyso-Gb_1_ analogs, as well as creatinine in a GD patient urine sample. (**B**) Ion chromatograms obtained by MRM mode of polycyclic lyso-Gb_1_ analogs in a GD patient sample. CPS: counts per second.

**Figure 4 diagnostics-12-01414-f004:**
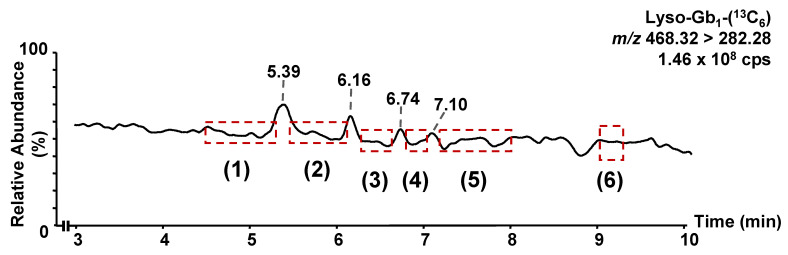
Post-column infusion of lyso-Gb_1_-(^13^C_6_) (1 µg/mL; 0.15 µL/min) during the injection of a healthy control plasma sample. The matrix effect in the elution regions of the biomarkers analyzed was estimated to be <15% for all analytes. In (**1**) elution region for lyso-Gb_1_ and lyso-Gb_1_-(^13^C_6_), as well as analogs −28, +14, +16 (A), and +30 Da. In (**2**) elution region for −12 (A), +2, +14, +16 (A), and +16 Da (B). In (**3**) elution region for polycyclic analog 390 Da, analogs −26, and −12 Da (B). In (**4**) elution region for analog +32 Da. In (**5**) polycyclic analogs 362 and 394 Da while polycyclic analog 366 Da elutes in the region (**6**).

**Figure 5 diagnostics-12-01414-f005:**
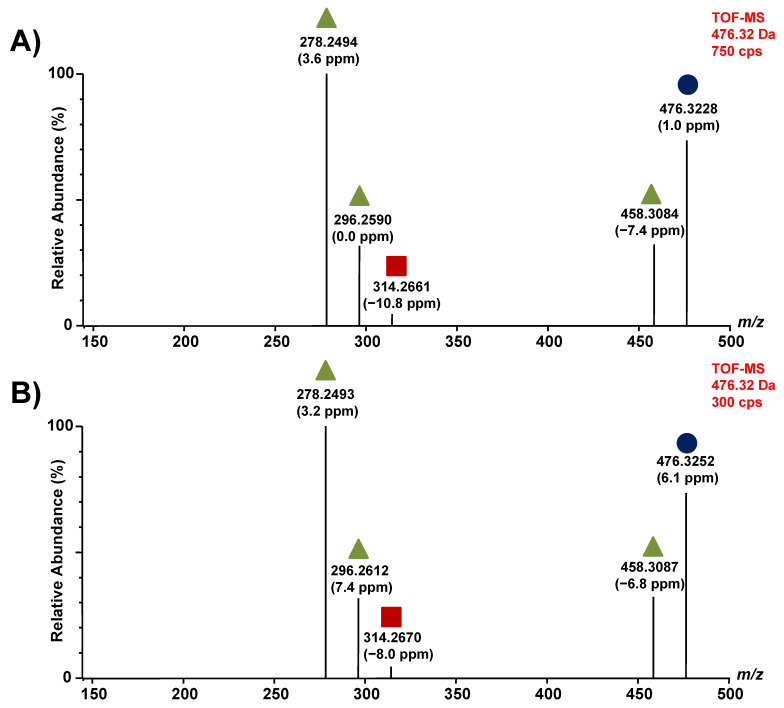
Fragmentation of lyso-Gb_1_ analogs +14 Da: (**A**) fragments obtained from the metabolite with a retention time of 5.30 min; (**B**) fragments obtained from the metabolite with a retention time of 5.53 min; in both (**A**,**B**), the precursor ion is indicated by a purple circle, a green triangle shows a loss of a water molecule, and a red square indicates a loss of a sugar unit. Parts per million (ppm) refers to the difference between experimentally measured masses (accurate mass) and theoretical masses (exact mass). CPS refers to the number of counts per second.

**Figure 6 diagnostics-12-01414-f006:**
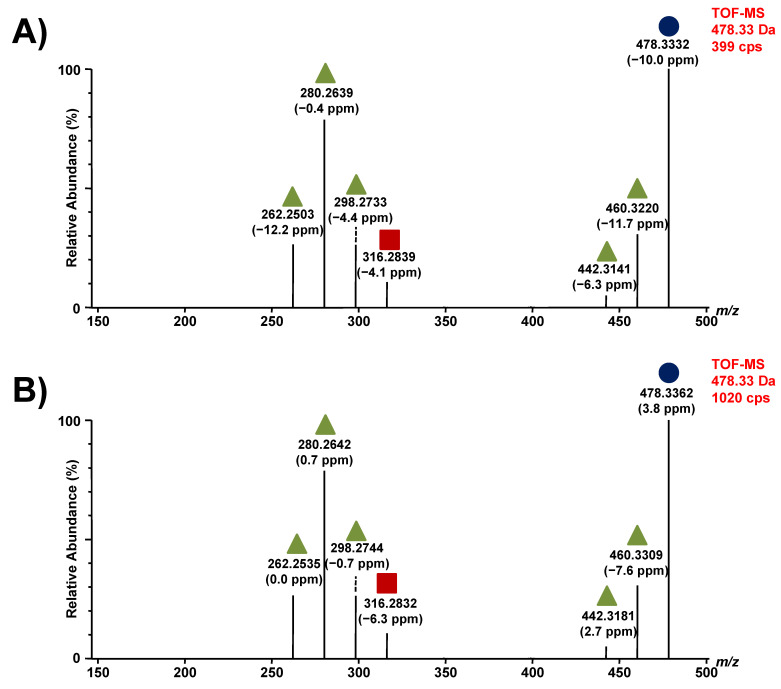
Fragmentation of lyso-Gb_1_ analogs +16 Da. In (**A**), fragments from the metabolite with a retention time of 5.43 min; In (**B**), fragments obtained from the metabolite with a retention time of 5.65 min; in both (**A**,**B**), the molecular ion is indicated by a purple circle, a green triangle shows a loss of a water molecule, and a red square indicates a loss of a sugar unit. CPS: counts per second. Parts per million (ppm) refers to the difference between experimentally measured masses (accurate mass) and theoretical masses (exact mass).

**Figure 7 diagnostics-12-01414-f007:**
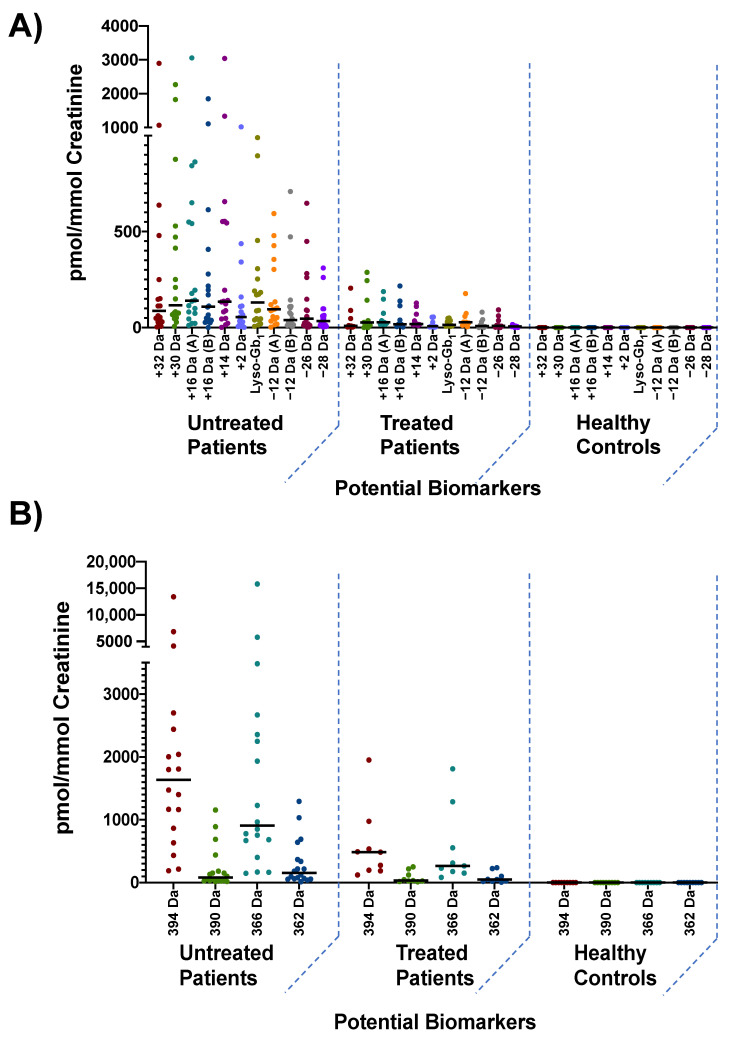
Distribution of: (**A**) lyso-Gb_1_ and analogs +32, +30, +16 (A) and (B), +14, +2, −12 (A) and (B), −26, as well −28 Da; and (**B**) Distribution of polycyclic analogs 394, 390, 366, and 362 Da in untreated (*n* = 18), treated (*n* = 9) and healthy controls (*n* = 9). The median concentration for each potential biomarker in indicated by a horizontal black bar.

**Figure 8 diagnostics-12-01414-f008:**
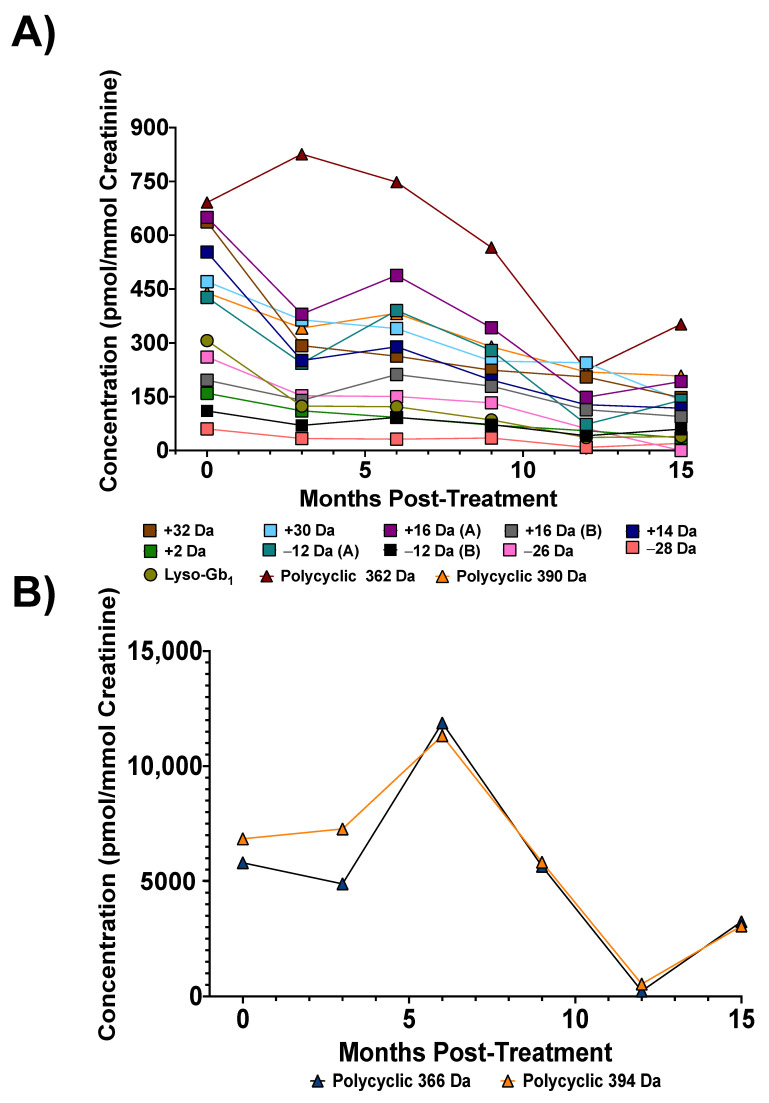
Biomarker concentration levels post-ERT treatment in a 55-year-old female GD type 1 patient with p.N370S/D409H mutations: (**A**) Concentration variation levels post-ERT for lyso-Gb_1_, lyso-Gb_1_ analogs +32, +30, +16 (A), +16 (B), +14, +2, −12 (A) and (B), −26, and −28 Da, as well as polycyclic analogs 390 and 362 Da; (**B**) Concentration variation levels post-ERT for polycyclic lyso-Gb_1_ 394 and 366 Da.

**Table 1 diagnostics-12-01414-t001:** Ultra-performance liquid chromatography coupled to tandem mass spectrometry (UPLC-MS/MS) method parameters.

**Parameters**	**Description**			
**Chromatographic parameters**				
HPLC Column	Halo HILIC 2.7 Advanced Materials Technology (Wilmington, DE, USA)			
Column Dimension	4.6 × 1500 mm			
Particle Size	2.7 μm			
Column Temperature	24 °C			
Weak Wash Solvent	ACN			
Strong Wash Solvent	ACN			
Injection Mode	Partial Loop			
Injection Volume	4 μL			
Mobile Phase A	94.5:2.5:2.5:0.5 ACN: MeOH: H_2_O: FA + 5 mM Amm. Form.			
Mobile Phase B	10:89.5:0.5 ACN: H_2_O: FA + 5 mM Amm. Form.			
Flow rate	0.85 mL/min			
Gradient (% Mobile Phase B)	0.0 → 9.5 min: 12.5%			
	9.5 → 11.0 min: 70.0%			
	11.0 → 12.0 min: 12.5%			
**Mass spectrometry parameters**				
Ionization Mode	Electrospray Ionization (ESI)			
Polarity	Positive			
Acquisition Mode	Multiple Reaction Monitoring (MRM)			
Capillary Voltage	3.2 kV			
Desolvation Temperature	550 °C			
Desolvation Gas Flow	750 L/h			
Cone Gas Flow	150 L/h			
Source Temperature	150 °C			
**Analytes**				
**Compound**	**Transitions (*m/z*)**	**Cone Voltage** **(V)**	**Collision Energy (V)**	**Dwell Time** **(s)**
Creatinine	144.07 > 44.05	10	5	0.200
Creatinine-(D_3_)	117.09 > 47.07	10	5	0.200
Lyso-Gb_1_ −28 Da	434.31 > 254.25	38	18	0.021
Lyso-Gb_1_ −26 Da	436.25 > 238.18	38	18	0.021
Lyso-Gb_1_ −12 Da	450.31 > 252.23	38	18	0.021
Lyso-Gb_1_	462.34 > 282.28	38	18	0.021
Lyso-Gb_1_ +2 Da	464.29 > 284.22	38	18	0.021
Lyso-Gb_1_ −(^13^C_6_)	468.36 > 282.28	38	18	0.021
Lyso-Gb_1_ +14 Da	476.36 > 296.26	38	18	0.021
Lyso-Gb_1_ +16 Da	478.34 > 280.26	38	18	0.021
Lyso-Gb_1_ +30 Da	492.32 > 294.24	38	18	0.021
Lyso-Gb_1_ +32 Da	494.33 > 296.26	38	18	0.021
Polycyclic Analog 362 Da	362.18 > 164.11	38	18	0.021
Polycyclic Analog 366 Da	366.18 > 168.10	38	18	0.021
Polycyclic Analog 390 Da	390.21 > 192.14	38	18	0.021
Polycyclic Analog 394 Da	394.21 > 196.13	38	18	0.021

**Table 2 diagnostics-12-01414-t002:** Method validation parameters using lyso-Gb_1_ and creatinine standards.

Validation Parameter	Results	
	Lyso-Gb_1_	Creatinine
Average intraday precision (RSD *) (*n* = 15)	2.9%	2.0%
Average intraday accuracy (Bias) (*n* = 15)	3.8%	3.0%
Average interday precision (RSD *) (*n* = 15)	5.3%	3.9%
Average interday accuracy (Bias) (*n* = 15)	4.2%	4.3%
Calibration curve ** (*n* = 5)	r^2^ > 0.998	r^2^ > 0.998
Stability *** at − 30 °C	At least a month	At least a month
Stability *** at 4 °C	7 Days	7 Days
Stability *** at room temperature	48 h	72 h
Stability in the sample organizer (10 °C)	48 h	48 h
Freeze/Thaw (3 cycles) (Bias)	7.6%	5.3%
Glassware adhesion (Bias)	4.2%	2.2%
Plasticware adhesion (Bias)	5.8%	3.1%

* RSD: Relative Standard Deviation. ** r^2^ refers to Pearson’s correlation coefficient. *** Stability is defined by less than 15% variability from baseline values.

**Table 3 diagnostics-12-01414-t003:** Method validation results for lyso-Gb_1_ analogs and polycyclic analogs.

Biomarker	Parameters						
	LOD (pM)	LOQ (pM)	Intraday Precision (%RSD)		Interday Precision (%RSD)		Recovery (%)
			LQC	HQC	LQC	HQC	
Lyso-Gb_1_ −28 Da	56	186	ND *	9.5	ND *	14.3	82.5
Lyso-Gb_1_ −26 Da	108	360	5.3	10.9	11.9	11.3	87.6
Lyso-Gb_1_ −12 Da	106	354	1.8	6.5	10.6	8.2	78.9
Lyso-Gb_1_ −12 Da (B)	98	328	11.2	5.3	14.8	11.3	81.3
Lyso-Gb_1_ +2 Da	101	337	7.8	6.9	12.2	8.5	82.3
Lyso-Gb_1_ +14 Da	84	283	6.2	7.6	8.5	7.2	72.1
Lyso-Gb_1_ +16 Da	105	351	6.9	2.1	15.9	11.1	78.9
Lyso-Gb_1_ +16 Da (B)	89	296	7.7	10.9	6.7	10.5	85.0
Lyso-Gb_1_ +30 Da	95	317	9.7	6.1	11.1	6.5	91.2
Lyso-Gb_1_ +32 Da	90	301	5.1	8.6	8.2	13.0	80.0
Polycyclic Analog 362 Da	105	305	3.3	5.6	6.9	4.7	98.3
Polycyclic Analog 366 Da	94	314	21.1	12.7	37.9	33.7	51.8
Polycyclic Analog 390 Da	109	364	5.2	4.2	21.6	15.3	82.8
Polycyclic Analog 394 Da	91	303	18.3	9.2	23.3	21.4	65.5

* ND: Not Detected.

**Table 4 diagnostics-12-01414-t004:** Clinical data for 55-year-old female GD type 1 patient post-ERT.

Months Post-ERT	Platelet Count (×10^3^ mm^3^)	Hemoglobin (g/dL)	Liver Volume (by MRI *)	Spleen Volume (by MRI *)
0	94	12.9	1666	976
3	105	13.3	-	-
6	136	13.0	1200	750
9	185	13.8	-	-
12	184	13.9	1302	570
15	124	14.5	-	-

* MRI: Magnetic Resonance Imaging.

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
