# Peer review of "Quantitation of a Urinary Profile of Biomarkers in Gaucher Disease Type 1 Patients Using Tandem Mass Spectrometry"

_diagnostics, 2022, doi:10.3390/diagnostics12061414_

Round 1

Reviewer 1 Report

Article carried out by Dr Menkovic et al, where the main objective is to develop and validate a tandem mass spectrometry urine quantitation method for LysoGb1 and analogs and poly cyclic LysoGb1 analogs and correlate with clinical manifestations of type 1 Gaucher disease. However, the article has serious defects:

  • In the introduction, citations are poorly selected: references 1 to 5 do not correspond to what is mentioned in the text. On the other hands citation number 8 mentions chaperones as a treatment in GD (and it is an unaccepted treatment) and citation 11 refers to GD type 3 (not GD type 1). Likewise, citations 12 and 13 are not appropriate to what is mentioned, and citation 17 does not mention Gaucher's disease.
  • The material and methods do not show the clinical characteristics or the scores of the patients. This should be strictly necessary if the investigation of new biomarkers is considered and limits the usefulness of the study.
  • In the results and discussion section, the authors continually mention citation 22, which corresponds to the same authorship. Can the authors explain the difference between this article and the one published in the Journal of Proteome Research (2022)?
  • The conclusions are not such, and the author is allowed to speculate with comments that should appear in the discussion. Furthermore, reading the article does not allow to conclude that the metabolites detected by the techniques used meet the necessary characteristics to be a valid Biomarker (Cox TM. Acta Pædiatrica 2005).
  • Finally, the bibliography should be revised: for example citation 27 is incomplete.

Author Response

Please see attached word document below.

Reviewer 2 Report

The authors has developed an analytical method for biomarkers of Gaucher disease. The biomarkers found by the authors seems to be promising for diagnosis. Therefore, the optimization and validation of the analytical method might be important. And the manuscript is well prepared. However, I have a few comments which should be clarified by the authors before publication.

  1. In section 2.8, adhesion of analytes to labwares were investigated. However, results were not described in 'Results and Discussion' section.
  2. In section 3.1, separation of lyso-Gb1 and psychosine was described. However, the authors already report the separation in previous paper (J. Proteome Res. 2022, 21, 5, 1321). Instead of explaining the reason, I would like to recommend to report the improvement from the previous paper.
  3. In table 3, LOD and LOQ of biomarkers were shown. How did the authors determine the LOD and LOQ without standard? 

Author Response

(The authors gave the same response as above.)
